# Estimation of Hydrological Components under Current and Future Climate Scenarios in Guder Catchment, Upper Abbay Basin, Ethiopia, Using the SWAT

Tewekel Melese Gemechu , Hongling Zhao, Shanshan Bao, Cidan Yangzong, Yingying Liu, Fengping Li and Hongyan Li *

Key Laboratory of Groundwater Resources and Environment, Jilin Provincial Key Laboratory of Water Resources and Environment, Jilin University, Changchun 130021, China; gemechu19@mails.jlu.edu.cn (T.M.G.); zhaohl19@mails.jlu.edu.cn (H.Z.); baoss16@mails.jlu.edu.cn (S.B.); aerlin321@163.com (C.Y.); liuyingying19@mails.jlu.edu.cn (Y.L.); fpli@jlu.edu.cn (F.L.)
* Correspondence: lihongyan@jlu.edu.cn; Tel.: +86-137-5625-7761

**Abstract:** Changes in hydrological cycles and water resources will certainly be a direct consequence of climate change, making the forecast of hydrological components essential for water resource assessment and management. This research was thus carried out to estimate water balance components and water yield under current and future climate change scenarios and trends in the Guder Catchment of the Upper Blue Nile, Ethiopia, using the soil and water assessment tool (SWAT). Hydrological modeling was efficaciously calibrated and validated using the SUFI-2 algorithm of the SWAT model. The results showed that water yield varied from 926 mm to 1340 mm per year (1986–2016). Regional climate model (RCM) data showed, under representative concentration pathways (RCP 8.5), that the precipitation will decrease by up to 14.4% relative to the baseline (1986–2016) precipitation of 1228 mm/year, while the air temperature will rise under RCP 8.5 by +4.4 °C in the period from 2057 to 2086, possibly reducing the future basin water yield output, suggesting that the RCP 8.5 prediction will be warmer than RCP 4.5. Under RCP 8.5, the total water yield from 2024 to 2086 may be reduced by 3.2 mm per year, and a significant trend was observed. Local government agencies can arrange projects to solve community water-related issues based on these findings.

**Keywords:** hydrological components; water yield; SWAT; RCM; RCP; Guder Catchment; Abbay Basin; Ethiopia

## 1. Introduction

Sustainable socio-economic development is built on the foundation of water resources [1,2]. The availability of water resources has diminished in recent decades, and some places of the world are experiencing increased water resource stress due to global climate change. Globally, the water demand will rise in future decades, mainly owing to the population increase and rising opulence; regionally, significant changes in agricultural water demand and in the impact on food availability, stability, access and utilization due to changes in water quantity and quality are projected as a result of climate change [3]. Water is one of the numerous vital concerns confronting Africa today and in the future. The water levels of lakes, rivers and groundwater also change over time [4]. The population at risk of increased water stress in Africa has been estimated to be 75–250 million in the 2020s and will rise to 350–600 million people by the 2050s. The impact of climate change on water resources across the continent is not uniform [5]. Using the IPCC's SRES scenarios from the HadCM320 climate model, the study by Arnell (2004) showed that runoff in the north and south of Africa is decreasing significantly, while runoff in eastern Africa and sections of semi-arid Sub-Saharan Africa, including Ethiopia, is expected to increase by 2050 [6]. Changes in hydrological cycles and water resources will certainly be a direct consequence of climate change [7].

One of Ethiopia's most vulnerable places to climate change is the Ethiopian portion of the Blue Nile (Abbay) river basin [8]. Ethiopia has twelve major watersheds. Basin studies were first conducted in the 1950s, and the Abik river Basin study was conducted in 1964, followed by studies of the north basin (Tekeze, Mereb-Gash and Guang) and the Wabi Shebele River Basin [9]. The Upper Blue Nile Basin in the highlands of Ethiopia plays a vital role in the Nile region's hydrology, accounting for more than 60% of the total flow of the Nile to Sudan and Egypt from the Upper Nile Basin (UNB) in Ethiopia [10]. According to a study from 2000 [11], the contribution of the Guder river to the Nile/Abbay river is 0.7%, and an estimation of the water productivity, since there are many challenges over water resources, was stated [12]. The HadCM3A2a and HadCM 3B2a SRES climate model outputs have been used to simulate and forecast climate at the local scale in the Guder Catchment [13].

The watershed modeling work has been concentrated on the representation of hydrological routes [14]. Arnold developed a SWAT model integrated with arc GIS at the USDA Agricultural Research Service (ARS) [15,16]. It was designed to predict long-term effects for wide ranges of soil-, land- and water-management circumstances, sedimentations and farming constituents [17–19]. The SWAT model was verified for the simulation of runoffs and sediment in the Abbay Basin, showing satisfying results and good performance in the watershed basin, including the Guder Catchment [20–23].

There is a limited number of meteorological and hydrological stations in the region, and the scarcity of essential climate and flow data for the Guder watershed reinforces the complications of assessing the hydrology. However, for water resource management, it is critical to anticipate dominating hydrological processes in data-scarce locations using existing data through semi-distributed hydrological models. Hydrological information is crucial for the better management of water resources to meet economic and environmental needs as it provides accurate information on water supply and quality for all users and helps to manage watersheds. Thus, studies of current water balances and current and future water yield may provide a more concrete basis for planning water projects and management. An essential component of water balance is water yield, which determines the availability of water for consumptive use (e.g., drinking or irrigation) or for in situ water supplies (e.g., water for hydropower or fisheries). In this study, we used the results of a state-of-the-art RCM formed from a combination of CMIP5 and a soil and water assessment tool (SWAT) hydrological model calibrated and validated using multiple spatially distributed gauge stations to estimate the water balance components under current and future climate change scenarios in the Guder Catchment of the Upper Abbay River Basin. Therefore, the objectives of this study were to estimate water balance components and water yield under current and future climate change scenarios and trend analysis in the Guder Catchment.

## 2. Methodology

### 2.1. Study Area Description

The Guder Basin is located in the Oromia region, Upper Blue Nile/Abbay Basin, Ethiopia, and located between 7°30′ to 9°30′ N latitude and 37°00′ to 39°00′ E longitude (Figure 1). The river flows into the Abbay river from south to north. The Abbay Basin (where the Guder Catchment is located) covers many hydroclimatic areas, from tropical wet to semi-arid and arid. Despite its proximity to the equator, the Guder sub-basin has a relatively mild climate due to its high elevation (1500–3000 M.a.s. l).

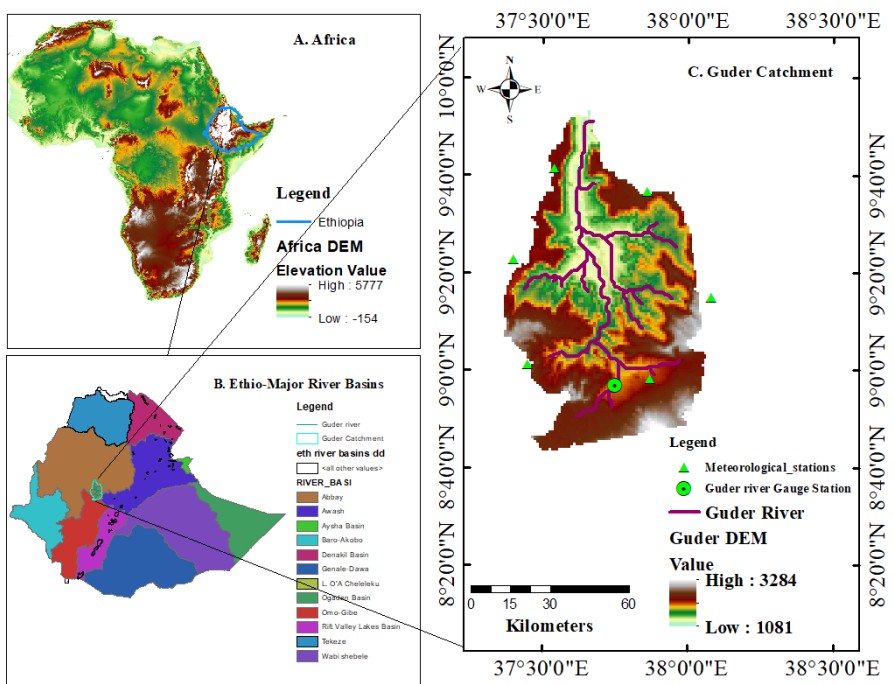

**Figure 1.** (**A**) Africa and (**B**) Ethio-major river basins and study area (Guder Catchment).

The annual average rainfall in the Guder Basin is 1228 mm and is categorized as a uni-modal rainfall through to the highest rainfall in July and August. The rainy period is from May to October, and the dry season is from November to April. The minimum and maximum air temperatures of the Guder Basin (1986–2016) were 16.6 °C and 23.7 °C, respectively, and the highest (April and March) and lowest air temperatures (July and August) were also recorded (see Figure 2).

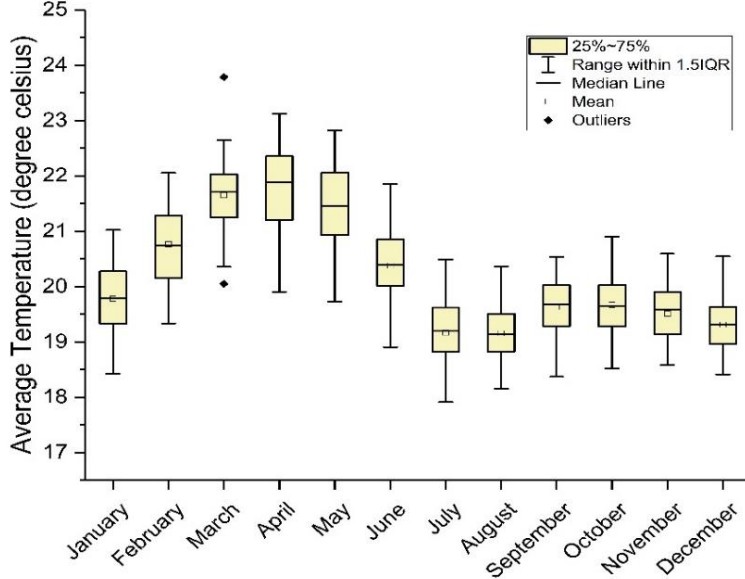

**Figure 2.** Mean monthly temperature of Guder Catchment (1986–2016).

### 2.2. Data Sources

A topography, land-use map, soil map and weather data are essential inputs for a SWAT simulation. The precipitation, minimum and maximum air temperature records are crucial to run the SWAT project [20]. These two meteorological records, daily rainfall and air temperature (1983–2016) from 6 stations, were collected from Ethiopian MoWIE,

the hydrology department at Guder River Basin stations, and data on land use soil were collected from the department of the GIS. The Shuttle Radar Topography Mission (SRTM) has access to the shapefile data of every country, and the Digital Elevation Model of Guder Catchment was downloaded from the Consortium for Spatial Information (CGIR-CSI) website (http://srtm.csi.cgiar.org/ (accessed on 25 January 2020)).

The daily streamflow/discharge records between 1983 and 2001 from the Guder river station were taken from the Ethiopian MoWIE. The Upper Abbay Basin region (including Guder Catchment) has a small number of meteorological and hydrological stations [21].

### 2.3. SWAT Model Principle and Setup

Before creating the SWAT project setup, the Digital Elevation Model shapefile was projected on an arc map with Ethiopian projection UTM Zone 37 N. The SWAT model uses HRU to identify the spatially different distribution of land use or land cover (Figure 3) to the soil type in the basin. HRUs are well defined independently for every sub-basin during the SWAT project building, centered on land use, soil type and slope in a definite sub-basin. The land use cover (Table 1) and soil shapefile of Guder Catchment (Figure 4) were overlapped and clipped to use for further hydrologic response unit definition.

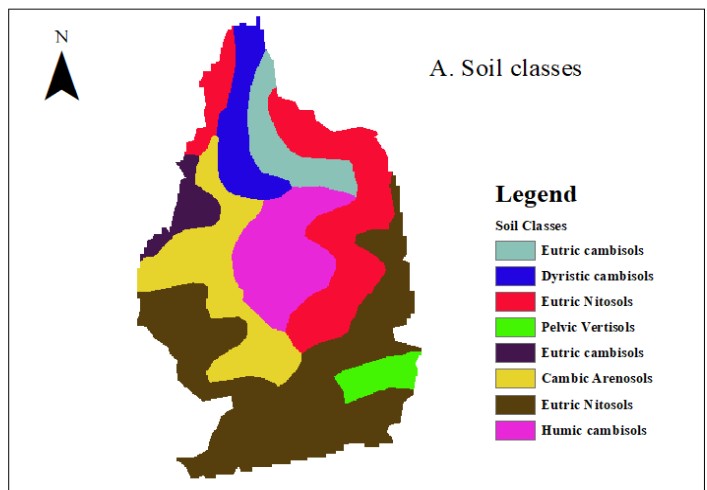

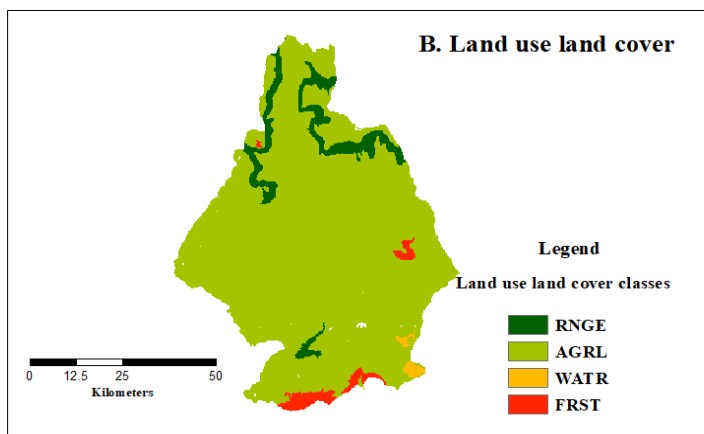

**Figure 3.** (**A**) Soil classes and (**B**) land use/land cover classifications.

**Table 1.** Land use coverage.

| Landuse | Area (km²) | Percent (%) |
|---|---|---|
| Range-Grasses (RNGE) | 411 | 7 |
| Agricultural Land-Generic (AGRL) | 5289.5 | 91.8 |
| Water (WATR) | 14 | 0.24 |
| Forest-Mixed (FRST) | 45.47 | 0.78 |
| Total | 5759 | 100 |

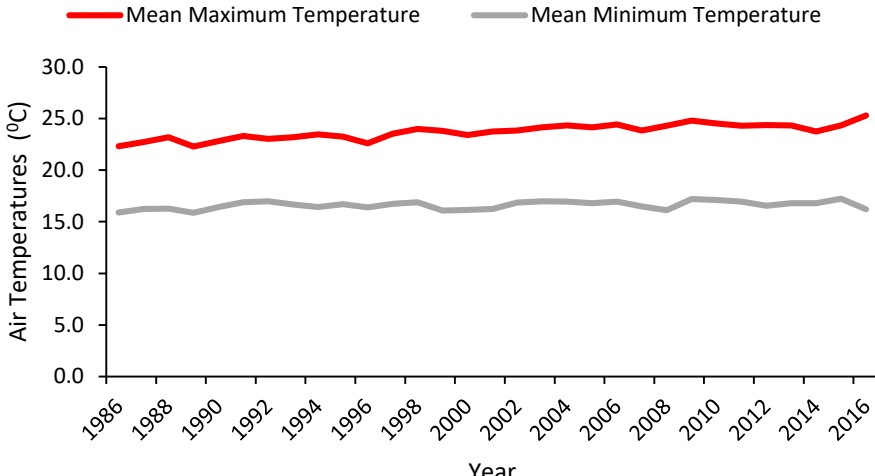

**Figure 4.** Mean maximum and minimum air temperature in Guder Catchment.

The Hydrologic Response Unit (HRU) definition in the Arc-SWAT aids in bringing the land use, soils and slope map to the SWAT project. The watershed that Arc-SWAT delineates in the watershed delineation section and land use and soil layers clipped from using the Arc Map on Arc GIS were overlapped. After re-allocation, finally, the whole watershed region in a Guder Catchment remained altered into Hydrologic Response Units with a total of 33 sub-basins.

Due to its effects on plant development and the flow of sediments, nutrients, pesticides and pathogens across the watershed region, water balance is the primary driving factor behind every process in the SWAT model [22]. In SWAT simulation, the equation of water balances in the base of the land phase hydrologic cycle that can be defined as the formula for the temporary water balance applied to the movement of water in the soil is described in the following equation:

$$SW_t = SW_o + \sum_{i=1}^{t}(R_{day} - Q_{surf} - E_a - W_{seep} - Q_{gw}) \tag{1}$$

where $SW_t$ is the final soil water content (mm); $SW_o$—the initial soil water content on the day $i$ (mm); $t$—the time (days); $R_{day}$—the amount of precipitation on a day $i$ (mm); $Q_{surf}$—the amount of surface runoff on the day $i$ (mm); $E_a$—the amount of evapotranspiration on the day $i$ (mm); $W_{seep}$—the amount of water entering the vadose zone from the soil profile on the day $i$ (mm); and $Q_{gw}$ is the amount of return flow on the day $i$ (mm).

SWAT deals with the Soil Conservation Service (SCS) curve number equation and the Green and Ampt infiltration methods [23]. SWAT uses three methods to estimate the potential evapotranspiration (PET): Priestly–Taylor, Penman–Monteith and Hargreaves methods [24]. These approaches' input data requirements are different. Solar radiation, air temperature, humidity and wind speed are required for the Penman–Monteith technique; solar radiation, air temperature and relative humidity are required for the Priestley–Taylor technique, and only air temperature is required for the Hargreaves method, which was chosen in this study.

SWAT-CUP (2012 Versions) is a computer program used to integrate various calibrations/uncertainty analysis programs to arc SWAT tools for calibration, validation and sensitivity analysis using the same interface. For this particular study, the Sequential Uncertainty Fitting version 2 (SUFI-2) algorithm was selected. The highest sensitive and significant stream discharge parameters were selected for modeling purposes at Guder Catchment. These parameters were collected from the research performed at the Abbay Basin and from the software developer manual [25,26].

Hydrological models are usually calibrated at longer times (monthly, seasonally, or annually) than their daily computing time (due to improved calibration, less processing requirements and the lack of reliable temporally fine data on the download observed (particularly in developing countries)) [27]. On the monthly time series for 19 years (1983–2001), including a three-year warm-up period, the calibration and validation of the model using the SUFI-2 algorithm program in the SWAT-Cup, applying the objective functions Nash–Sutcliffe Efficiency (NSE) and $R^2$, were performed. The streamflow data from 1986 to 1997 were used for calibration, and from 1998 to 2001 were used for validation. NSE [18] remains a commonly used and possibly reliable statistic for evaluating the goodness of fit of the hydrological model, and the ranges of the value are from 1 (best) to $(-\infty)$ negative infinity. PBIAS was used to measure whether the simulated value was more or less than that of the measured data. The optimized PBIAS value is zero, and low values show modified models. Positive values show the model's overestimation, while the model's underestimation can be shown from the negative values.

### 2.4. Trend Analysis

The nonparametric (MK) Mann–Kendall and Sen's slope estimates method was used in this analysis to evaluate and detect patterns in statistical time series databases based on annual results. The XLSTAT tools (XLSTAT 2019) were installed as an Excel extension, and MK on MATLAB provided an efficient, complete and user-friendly data analysis and statistical solution. The following formulas were used in MK-Test and SEN's at MATLAB workspace to obtain and correct patterns in annual results.

$$xi = f(ti) + \varepsilon i \tag{2}$$

where f(ti) is a continuously monotonous increase or decrease in time; the $\varepsilon i$ residues are considered to be of the same zero-mean distribution.

H1, as an alternative hypothesis, states that there is a rising or declining monotonic trend, is pitted against the null hypothesis of no trend, Ho, in which the observations xi are randomly ordered in time:

$$S = \sum_{k=1}^{n-1} \sum_{j=k+1}^{n} sgm(X_j - X_k) \tag{3}$$

where $x_j$ and $x_k$ are the annual values in years' $j$ and $k$, $j > k$, respectively.

$$sgm(X_j - X_k) = \left\{ \begin{array}{ll} 1 & if\ X_j - X_k > 0 \\ 0 & if\ X_j - X_k = 0 \\ -1 & if\ X_j X_k < 0 \end{array} \right\} \tag{4}$$

$$VAR(S) = \frac{1}{18} \left[ n(n-1)(2n+5) - \sum_{p=1}^{q} t_p(t_p - 1)(2t_p + 5) \right] \tag{5}$$

The $p$th group denotes the number of data values with $t_p$ and the number of bound groups with $q$.

The true path of a current trend is determined by the nonparametric method of Sen (as the change each year). This implies that in equation five (5), the above f(t) equals the following:

$$f(t) = Q_i + B \tag{6}$$

where *Q* is the slope and B is a constant.

$$Q_i = \frac{X_j - X_k}{j - k} \tag{7}$$

where *j* is greater than *k* if the time series contains *n* values $x_j$, then N = $n (n - 1)/2$ slope estimates $Q_i$. The median of these N $Q_i$ values is Sen's estimator of the slope. The $Q_i$ values are arranged from smallest to largest in Sen's estimator, with t = year first.

*2.5. Climate Scenarios and Bias Correction*

The World Climate Research Program's Organized Coordinated Regional Climate Downscaling Experiment (CORDEX) program is currently providing an opportunity to develop regional climate predictions that could be used with high-resolution on a regional scale to assess possible climate change impacts. The study selected RCP 4.5 (stabilization scenario) reductions in energy consumption and RCP 8.5 (high emission scenario) indicative of high emission scenarios from the four RCP scenarios used in CMIP5, leading to radiative forcing values. Ethiopia's RCM in CORDEX was statistically downscaled and utilized in hydrological modeling as input data. For use in climate impact modeling, this data set includes downscaled data for different forecasts with a spatial resolution of $0.5° * 0.5°$ (approximately 50 * 50 km). The historical database spans from 1 January 1976 to 31 December 2005, while future climate prediction scenario data (RCP 4.5 and RCP 8.5) span from 1 January 2021 to 31 December 2100. Each *.nc _le contains one climate variable (i.e., pr, tasmax and tasmin) that spans five years. Due to structural model errors and discretization and spatial averaging within grid cells, simulated climate data cannot be used as direct inputs to hydrologic models. On a standard time scale, to minimize the disparity between measured and modeled climate change, bias correction techniques are used so that hydrological simulations based on corrected climate simulation data comply with the simulations based on climate observations [28].

The CMhyd tool was conceived to offer climate information simulated that could be used to decide where gages should be placed in a watershed model. As a result, climate model data for each gaging station position should be obtained and bias-corrected [25]. The netCDF metadata are used by CMhyd to convert the precipitation and air temperature data into millimeters and degrees Celsius, respectively, and the climate model grid cells overlaying the gaging sites. Finally, CMhyd reads the netCDF file to extract time series from linked grid cells [29].

Distribution mapping of precipitation and air temperature is intended to compare the distribution function of the RCM expected outcome values with the distribution function measured. This can be achieved by creating a transfer feature that changes the precipitation and air temperature distribution. According to [28], distribution mapping is the most effective correction method. It corrects most statistical features and has the smallest deviation ranges when paired with the best overall mean fit. Thus, for this study, using CMhyd instruments for a selected position at the Guder Catchment, the data from the models were extracted and used for further analysis.

## 3. Hydrological and Climatic Characteristics in Guder Catchment

The distribution of precipitation over the Guder Catchment from 1983 to 2016 was analyzed. While the Kriging technique is the preeminent indirect estimate for a given rainfall network, the yearly average precipitation in the basin reflects the best annual precipitation [30]. The amount of rainfall at the basin and sub-basin levels varies by location. The rainfall distribution from a set of data on sites provides ways to estimate



point (pole) and polar (region) values on the map. The spatial average rainfall distribution in the basin was calculated using the data recorded from 6 rain gauges.

Hydrographs can be used to show flow differences and to identify high and low flow times. Climate change may be accelerating global warming, as global warming will increase steam levels. The calculation of annual distribution with an inhomogeneity coefficient can be used for runoff and precipitation or other elements, and it is a feasible expression. A standard method for analyzing the annual variation of runoff is to use an uneven distribution of runoff coefficients throughout the year. The greater the value, the more significant the difference in monthly runoff during the year and the higher the uneven distribution of runoff throughout the year. The value of the coefficient of uneven distribution during the year is greater than or equal to zero and less than 1, as follows:

$$C_u = \sigma/\overline{R}, \quad \sigma = \frac{1}{12}\sqrt{\sum_{i=1}^{12}(R_i - \overline{R})^2}, \quad \overline{R} = \frac{1}{12}\sum_{i=1}^{12} R_i \tag{8}$$

Among them, $C_u$ is the coefficient of uneven distribution, $R_i$ is the monthly runoff and $\overline{R}$ is the average monthly runoff throughout the year.

Accordingly, the uneven monthly distribution of parameter coefficients (Table 2) and uneven analysis of the yearly flow production process (Table 3) have been described.

**Table 2.** Monthly uneven distribution of parameter coefficients.

| Value | Surface Runoff | Rainfall |
|---|---|---|
| $C_u$ | 0.29 | 0.27 |

**Table 3.** Uneven analysis of yearly flow production process.

|  | Surface Runoff | Rainfall |
|---|---|---|
| Max/Min | 1.79 | 1.46 |
| Max/Mean | 1.36 | 1.26 |
| Min/Mean | 0.76 | 0.86 |
| Cu | 0.21 | 0.18 |

*3.1. Temporal Characteristics*

3.1.1. Annual Characteristics

(A) Rainfall

As stated in above in Section 2.1, the maximum occurrence of precipitation is in July and August. The rainy period is from May to September, while the dry period is between October and April. Accordingly, in this research, July is also the month in which the highest precipitation of an average with (264.9 mm) was recorded, and December is the least dry month in which the slight rainfall with an average of (7.89 mm) was recorded (1986–2016) (Figure 5).

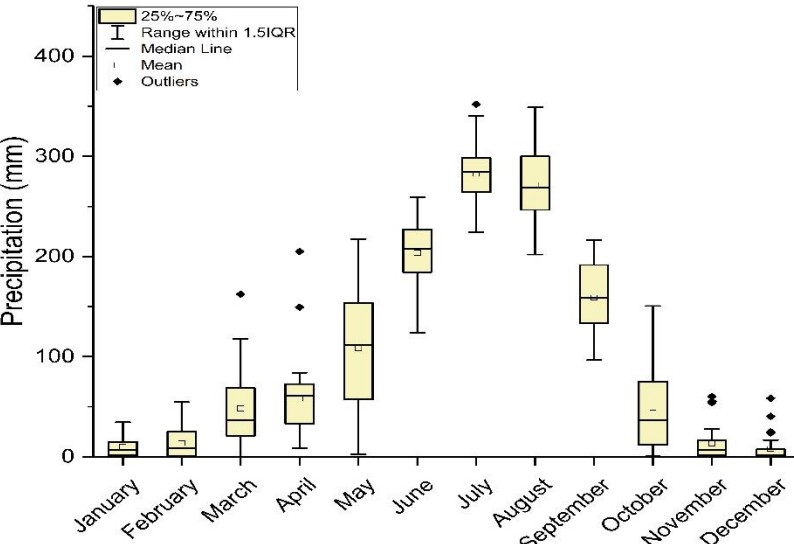

**Figure 5.** Annual average precipitation of Guder Catchment (1986–2016).

(B)    Surface Runoff

Surface runoff is water that flows from the ground, rainwater and other sources and is a significant integral part of the water cycle. The monthly average runoff over an area generated in the year 1986–2016 is described in Figure 6. The maximum runoff was in July and August, 66.27 mm and 60.78 mm, respectively. Additionally, the minimum runoff was between November and January (1.33 mm–2.96 mm). The runoff and rainfall in the Guder Catchment have a direct relationship. This may have resulted from the catchment's elevation, soil infiltration capacity, land cover and topography.

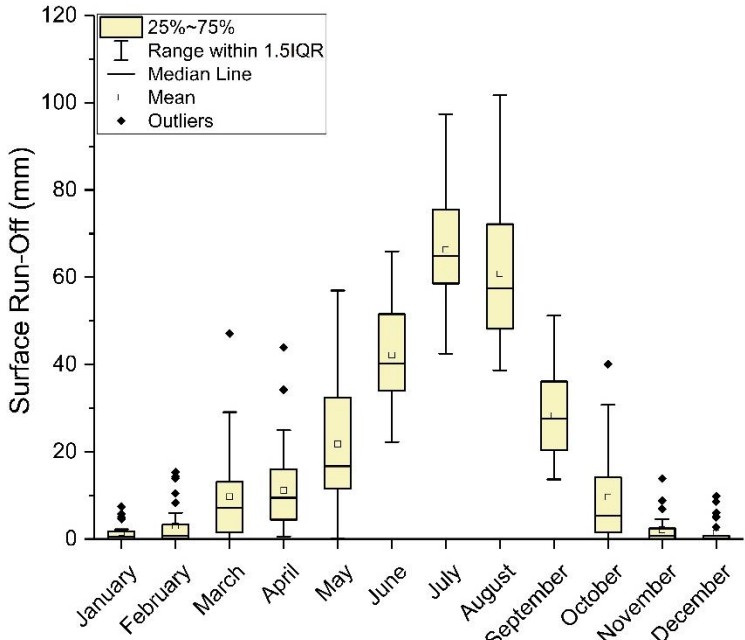

**Figure 6.** Mean monthly surface runoff in Guder Catchment (1986–2016).

The greater the value, the more significant the difference in monthly runoff during the year and the higher the uneven distribution of runoff throughout the year. The value of the coefficient of uneven distribution during the year is greater than or equal to zero.

### 3.1.2. Interannual Characteristics

(A)   Rainfall

According to this study, the highest rainfall (1469.9 mm) was recorded in 1992, and the minimum precipitation was recorded in 2002 (1003.8 mm) in the Guder Catchment between 1986 and 2016. However, there are no significant trends (see Figure 7).

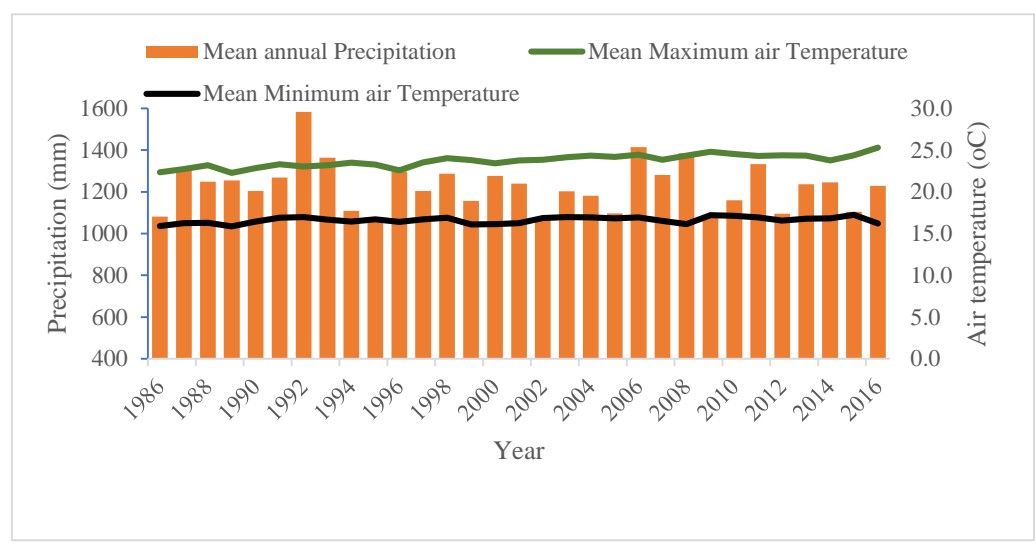

**Figure 7.** Yearly average precipitation and air temperature of Guder Catchment (1986–2016).

(B)   Surface Runoff

As a result of the highest and lowest precipitation in 1992 and 2002, respectively, the runoff showed the same change. And the highest and lowest runoff, 350.2 mm (1992) and 195.7 mm (1995) were generated, respectively, from the Guder Catchment area. From Figure 8, we can understand that, the surface runoff and precipitation has a direct relationship.

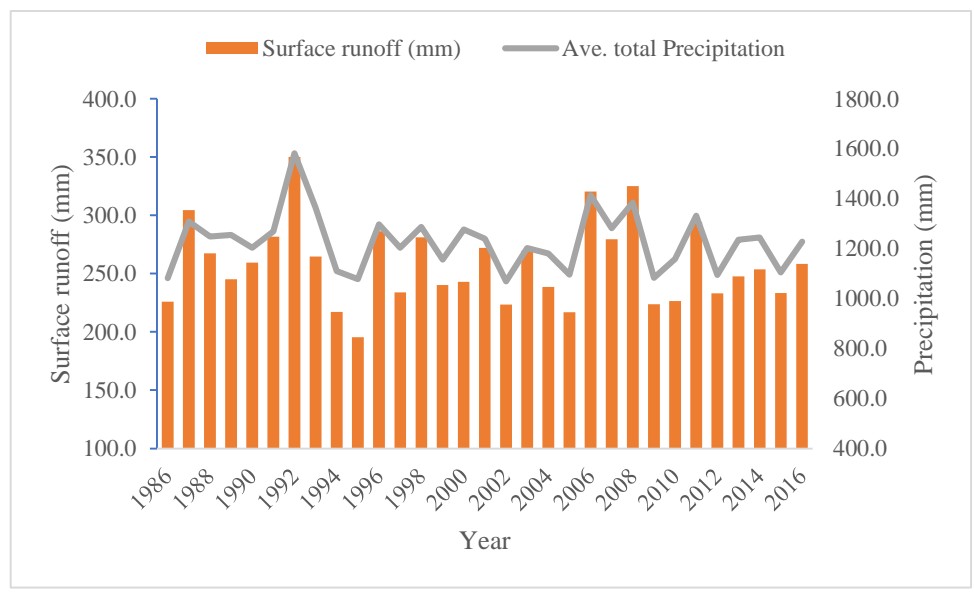

**Figure 8.** Yearly surface runoff and total precipitation relationships (1986–2016).

### 3.2. Spatial Distribution of Rainfall

The distribution of precipitation over every sub-basin in the Guder Catchment from 1983 to 2016 was analyzed. It was nearly evenly distributed from the upper sub-basin to

the lower sub-basin, as shown in Figure 9. The average annual rainfall at the southern end of the basin is 1505.7 mm, while the lowest annual rainfall in the northeast is 905.34 mm.

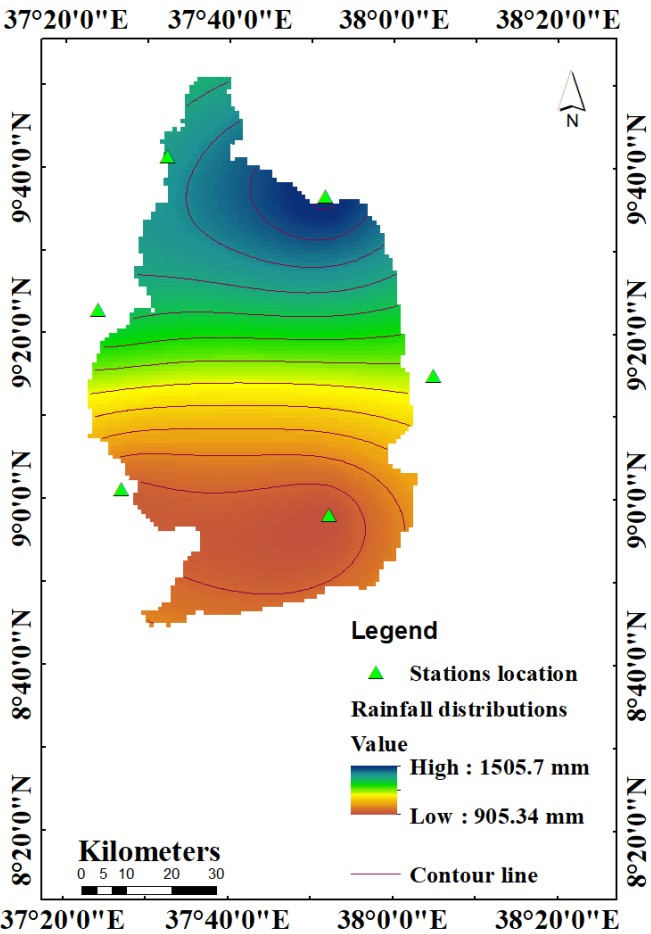

**Figure 9.** Spatial distribution of rainfall in Guder Catchment (1983–2016).

Although there is a positive relationship between rainfall and elevation in Ethiopia [31], when we consider it from a local perspective, the relationship is not consistent; some locations may negatively impact elevation on rainfall. Except for the western lowlands, where high rainfall, places with intermediate and higher altitudes (above 1500 m) receive significantly more rain [32].

## 4. Results

### 4.1. Model Calibration and Validation

The streamflow/discharge data used were from the Guder river gauge station to calibrate and validate the SWAT model. The streamflow calibration results showed acceptable agreement in which the values of NS and $R^2$ were 0.70 and 0.71, respectively. The observed and simulated discharge graph during calibration (Figure 10) and validation (Figure 11) shows good agreement. The validation of the model was performed with four years of discharge, from 1998 to 2001. Improved NS and $R^2$ were found at 0.86 and 0.85, respectively; generally, there was an acceptable fit between the measured and simulated output. This shows that the SWAT model can work in the simulations of hydrological components (Table 4). At the Guder Catchment, the top three most sensitive and significant flow parameters were V__GWQMN (threshold depth of groundwater in the shallow aquifer mandatory for return flow to occur (mm)), V__CH_K2 (effective Channel hydraulic conductivity (mm/h)) and R__CN2. (See Table 5).

The above results have good agreement with the results assessed by [18] on the runoff sediment yield modeling in this watershed in which the values of the simulated runoff during calibration and validation were 13.48 m$^3$/s and 9.54 m$^3$/s, respectively.

The hydrograph of the calibration and validation period of the observed and simulated flow in the monthly estimation shows that the model somewhat underestimates some of the month's peak flows while overestimating others. This could be due to the quality of the weather or flow data utilized as model input.

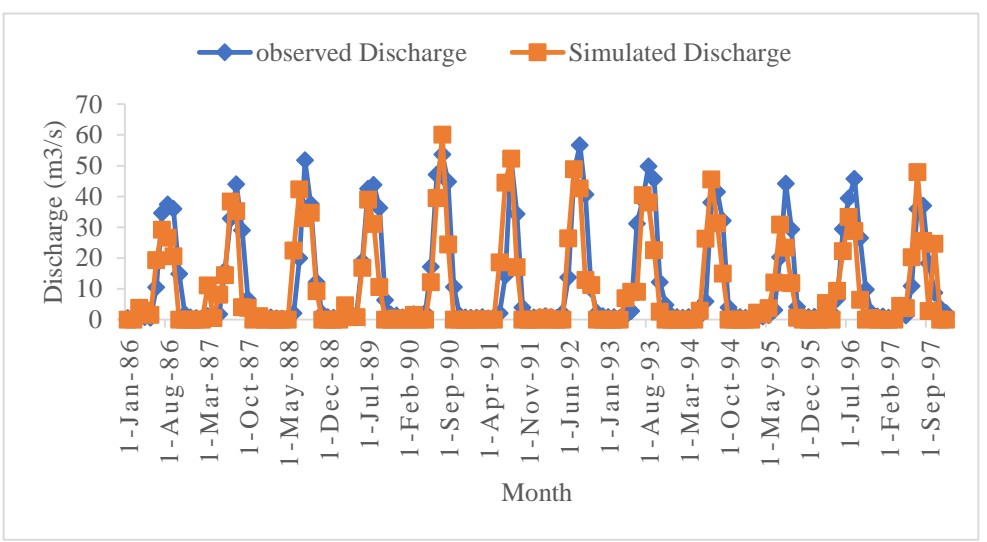

**Figure 10.** Calibration results of average monthly observed and simulated flow hydrograph (1986–1997).

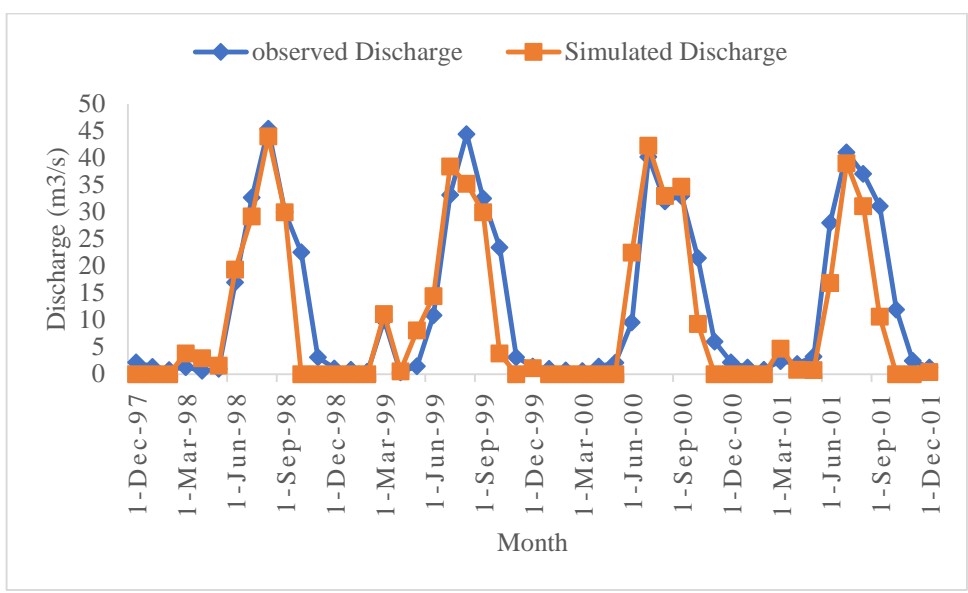

**Figure 11.** Validation results of average monthly observed and simulated flow hydrograph (1998–2001).

**Table 4.** Calibration and validation of monthly discharge.

|  | NS | R$^2$ | Average Simulated (M$^3$/S) | Average Observed (M$^3$/S) | PBIAS (%) |
|---|---|---|---|---|---|
| Calibration | 0.70 | 0.71 | 10.09 | 12.14 | 16.89 |
| Validation | 0.86 | 0.85 | 9.73 | 10.81 | 9.99 |

**Table 5.** Sensitivity analysis table.

| Sensitivity Rank | Name of Parameters | Fitted Value | t-Stat | *p*-Value |
|---|---|---|---|---|
| 1 | V__GWQMN.gw | 36.18 | 2.02 | 0.05 |
| 2 | V-CH_K2.rte | 31.5 | 1.98 | 0.05 |
| 3 | R__CN2.mgt | 38.79 | 1.92 | 0.06 |
| 4 | R__SOL_AWC (.).sol | 0.42 | −1.67 | 0.10 |
| 5 | V__ALPHA_BNK.rte | −13 | −1.5 | 0.14 |
| 6 | V__ESCO.hru | 0.59 | −1.40 | 0.16 |
| 7 | V__GW_DELAY.gw | 0.5 | −1.05 | 0.29 |
| 8 | R_SOL_K.sol | 12.25 | −0.98 | 0.33 |
| 9 | V__ALPHA_BF.gw | 0.84 | 0.85 | 0.40 |
| 10 | V__SFTMP.bsn | −10.2 | 0.70 | 0.48 |
| 11 | V_GW_REVAP | 0.07 | 0.63 | 0.52 |
| 12 | V__CH_N2.rte | 0.29 | −0.36 | 0.71 |
| 13 | R__SOL_BD (...).sol | 0.6 | 0.027 | 0.97 |

Note: 1. Threshold depth of water in the shallow aquifer; 2. effective hydraulic conductivity; 3. SCS runoff curve number; 4. available water capacity; 5. baseflow alpha factor for bank storage; 6. soil evaporation compensation factor; 7. groundwater delay; 8. saturated hydraulic conductivity; 9. baseflow alpha-factor; 10. snowfall temperature; 11. groundwater "revap" coefficient; 12. manning's "*n*" value; 13. moist bulk density.

*4.2. Water Balances*

The SWAT model estimates water balance components for semi-sub-divided model results for the Guder Basin (Table 6).

**Table 6.** Monthly main hydrological components of Guder Catchment (1986–2016).

| Month | Precipitation | Surface Runoff | Lateral Flow | ET | Percolation to the Shallow Aquifer | Water Yield |
|---|---|---|---|---|---|---|
| January | 11.03 | 1.5 | 0.4 | 3.3 | 6.1 | 21.6 |
| February | 15.17 | 2.96 | 0.4 | 4.2 | 8.1 | 14.2 |
| March | 47.68 | 9.7 | 1.3 | 7.7 | 27.1 | 22.8 |
| April | 58.00 | 11.13 | 1.8 | 9.8 | 34.5 | 32.5 |
| May | 104.41 | 21.76 | 3.6 | 11.8 | 65.5 | 57.4 |
| June | 192.42 | 42.14 | 6.8 | 11.7 | 129.3 | 102.3 |
| July | 264.90 | 66.27 | 9.9 | 15.6 | 172.0 | 174 |
| August | 253.89 | 60.78 | 10.2 | 14.2 | 168.4 | 207.5 |
| September | 147.38 | 28.27 | 6.9 | 11.6 | 101.6 | 177.7 |
| October | 45.27 | 9.63 | 2.9 | 5.6 | 29.8 | 131 |
| November | 12.48 | 2.09 | 1.0 | 3.7 | 7.2 | 73.3 |
| December | 7.89 | 1.33 | 0.5 | 2.9 | 4.2 | 40.1 |
| ANNUAL | 1228 | 257.56 | 45.6 | 102.1 | 753.8 | 1054.4 |

Note: For all components, unit is in mm.

Generally, the annual and monthly averages of hydrologic watershed components shows that rainfall are directly related to each water balance component. SWAT showed a tremendous ability to analyze and examine different water balance units. Hydrological watershed components are derived from various atmospheric variables as part of the SWAT model inputs. From the total rainfall, 9% was evaporated as evapotranspiration. The contribution of precipitation to percolation in the Guder Catchment is high (65%), and in contrast, it is low (16%) in the Ribb watershed near Lake Tana, Abbay Basin. This (closeness to the lake) may impact evapotranspiration in addition to different factors related to the geological characteristics and the potential groundwater of the area. Additionally, the contribution of surface runoff to precipitation is about 20%, and this contribution was related to the result from the Ribb watershed, Abbay Basin, Ethiopia [33]. The ratio of the total flow, base flow and ground flow was also estimated in the Guder Basin (Table 7).

This paper analyzed the water balance under different land-use types. The highest runoff was generated from agricultural lands, and the highest potential groundwater existed in range lands.

**Table 7.** Water balance ratios.

| Component | Base Flow/Total Flow | Runoff/Total Flow | Streamflow/ Precipitation | Percolation/ Precipitation | Deep Recharge/ Precipitation | Evapotranspiration/ Precipitation |
|---|---|---|---|---|---|---|
| Ratio | 0.75 | 0.25 | 0.88 | 0.65 | 0.03 | 0.09 |

### 4.3. Climatic Projections under RCP Scenarios

In the 2030s (2024–2053) and 2080s (2057–2086) of the projected reginal climate model at the Guder Catchment, the average annual air temperature may rise in both RCP scenarios. Maximum air temperature will slightly increase throughout the year overall in future periods. The mean maximum air temperature may increase under RCP 8.5 as (+4.4 °C) in the period of 2057–2086.

Water balance components are evaluated and summarized in Table 8 below based on the RCP climate scenarios. In this study, future changes in land use were assumed to be constant to examine the effects of climate variables on change, keeping all other factors constant. Under RCP scenarios, the Table 8 below summarizes the response of the hydrological components of the basin to an average annual level and percent changes in the basin (precipitation, evapotranspiration and total water yield).

**Table 8.** Change in water balance components.

| Scenario | Period | P (mm) | ET (mm) | WYLD (mm) |
|---|---|---|---|---|
| Baseline | 1986–2016 | 1228 | 104.2 | 1054.7 |
| RCP 4.5 | 2024–2053 | 1113 (−9.4%) | 87.0 (−16.5%) | 1040.7 (−1.3%) |
| | 2057–2086 | 1117 (−9.1%) | 86.4 (−17.1%) | 1040.4 (−1.4%) |
| RCP 8.5 | 2024–2053 | 1139 (−7.3%) | 88.7 (−14.9%) | 1048.7 (−0.6%) |
| | 2057–2086 | 1051.7 (−14.4%) | 87.9 (−15.7%) | 960.6 (−8.9%) |

P = precipitation; ET = evapotranspiration; WYLD = the total amount of water yield (overland flow + lateral flow + percolation + deep recharge).

From the above table, almost all water balance components showed minor reductions with a small difference from the baseline (1986–2016) to the first projection period (2024–2053) and 2nd projection period (2057–2086) under RCP 4.5 scenarios. By the 2080s (2057–2086), the reduction in precipitation, evapotranspiration and water yield may reach 9.5%, 15.7% and 8.9% (110.9 mm, 16.3 mm and 94.1 mm), respectively, in the RCP 8.5 scenario. The assumption given [4] for the air temperature rise by 2100, relative to pre-industrial air temperatures, is given by the RCP 8.5 pathway, which may impact the decrement of these parameters. In this study, the air temperature had a high impact due to its increment from 1.7 to 2.4°C under RCP 4.5 and 2.7°C to 4.4°C under RCP 8.5 from baseline to both future projections periods, 2024–2053 and 2057–2086. The average monthly water yield shows that the average water yield generated by each hydrological response unit in the catchment is varied and could be lower because of lower rainfall in the dry season. Whereas, due to high precipitation, water yield produced by HRU in the catchment could be very high during the rainy season (Figure 12).

The most precise method for measuring water yield is to calculate the value of each term in the water balance equation. According to the following Figure 13, the changes are negative, except during the rainy season (July–October). This means that the maximum water yield decline can be seen in June up to 32% under RCP 8.5 and 27% under RCP 4.5 scenarios from the baseline. However, water yield will increase in the rainy season and in August under RCP 4.5 (28%). In October, RCP 8.5, the maximum water yield increment change from the baseline might be (6%).

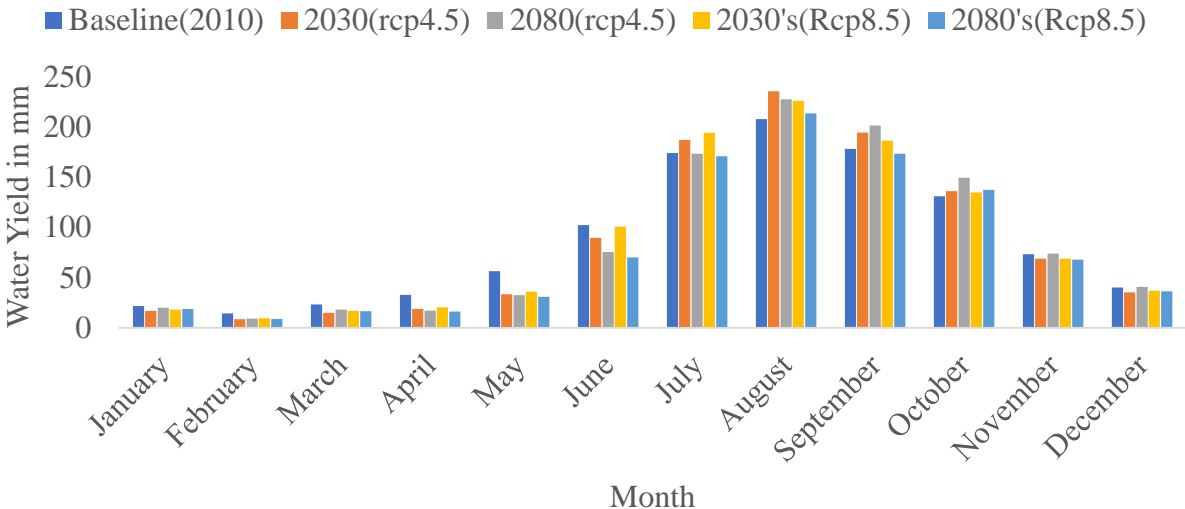

**Figure 12.** The monthly difference in water yield.

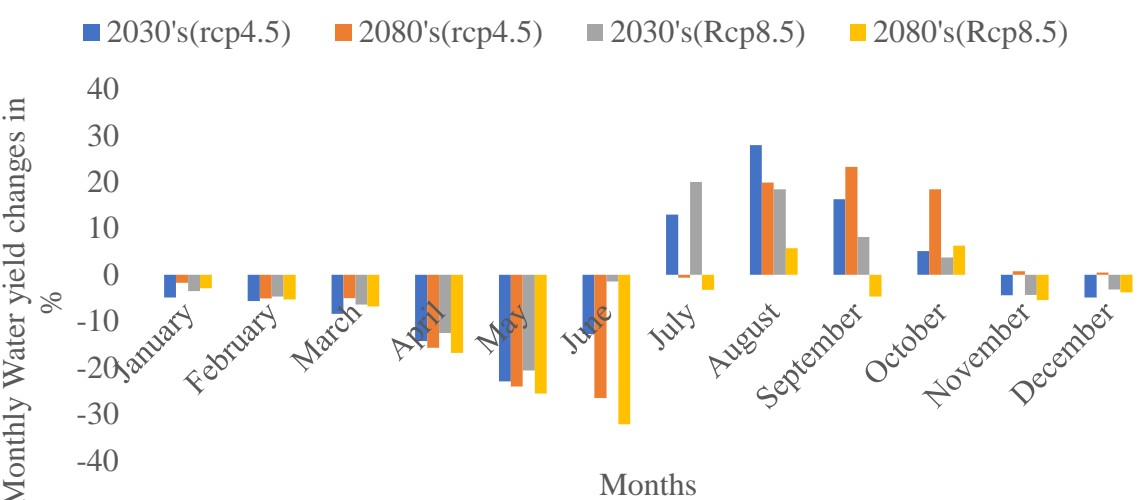

**Figure 13.** Monthly water yield changes.

### 4.4. Trends of Simulated Future Water Yield

In addition to the baseline, under the future RCP 4.5 and 8.5 scenarios, the change in water yield simulated by the model SWAT was determined. Accordingly, under RCP 8.5 from 2024 to 2086, the total water yield may be decreased by 3.2 mm per year (Table 9), and a significant downward trend was observed (Figure 14).

**Table 9.** Water yield trends in Mann–Kendall test and Sen's slop results for future scenario from 2024 to 2086.

| Series\Test | Mean | LCL | UCL | Z Statistic | Slope Estimate | Trend Significance |
|---|---|---|---|---|---|---|
| RCP 4.5 | 1040.5 | −3 | 3.3 | 0.2 | 0.37 | Not Significant |
| RCP 8.5 | 1004.6 | −7 | 0.4 | −2.2 | −3.2 | Downward trend |

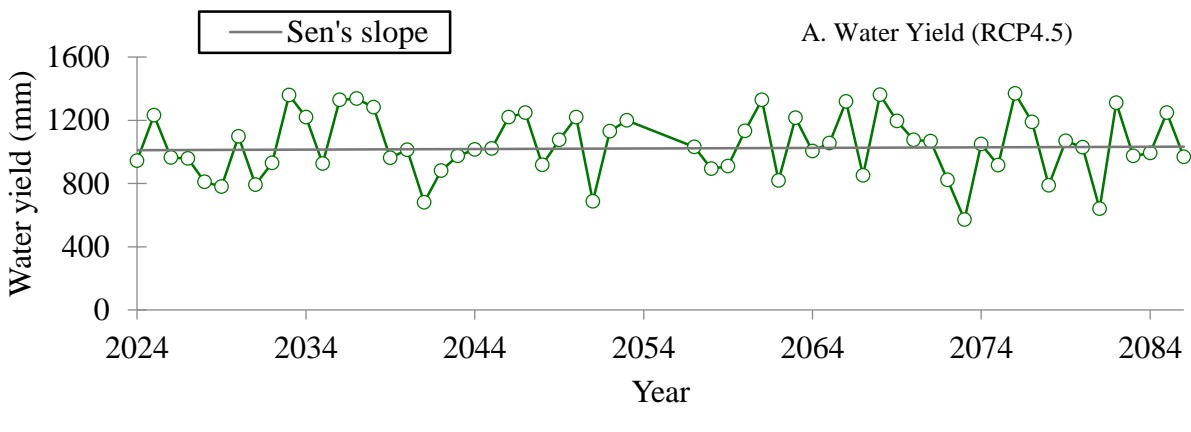

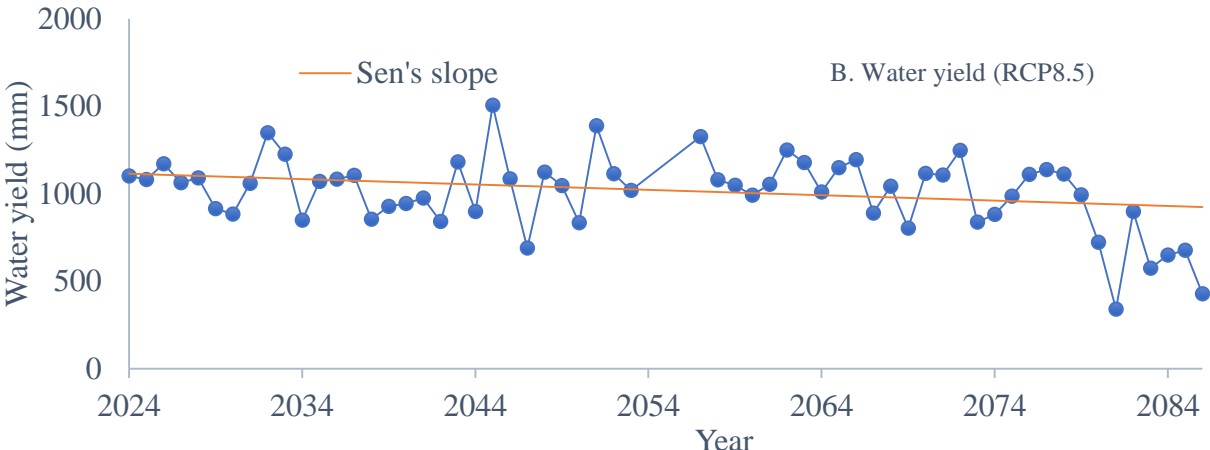

**Figure 14.** Simulated water yield trends under (**A**) RCP 4.5 and (**B**) RCP 8.5.

## 5. Discussion

The streamflow calibration results showed an acceptable agreement. There was an acceptable fit between the measured and simulated output. The SWAT model can be used in the simulations of hydrological components. The annual and monthly averages of hydrologic watershed components show that rainfall is directly related to each water balance component. The average annual rainfall at the southern end of the basin is 1505.7 mm, while the lowest annual rainfall in the northeast is 905.34 mm. Rainfall generally decreases from the southwest to the northeast with an elevation decrement. This temporal and spatial distribution of rainfall changes is essential for hydrological analysis, planning and water resource management. Evapotranspiration (ET) is a crucial component of the soil–water balance, which influences agricultural yield potential. Although influenced by several factors, such as soil air temperature, soil moisture and vapor pressure gradients, ET is nearly identical for a given soil texture class and agroclimatic conditions. This study showed a reduction in evapotranspiration under all RCP potential scenarios. Under both future scenarios, evapotranspiration can show a reduction of 14–17%.

As stated by [34], an increase in land use with tree cover can decrease water yield (surface runoff, groundwater flow and lateral flow). Surface runoff, lateral flow, near-surface aquifer and deep aquifer are all components of water yield. According to the assessment conducted in the Upper Abbay Basin [35], using the GCM/SDSM-downscaling method, water yield showed a reduction, including various water balance components. Additionally, in this particular study, water yield was projected to be decreased from 0.6% to 9% under both RCP 4.5 and RCP 8.5. This may be mainly due to the possible precipitation decline anticipated by the RCC scenarios and the increase in mean air temperature. According to [33], 70% of annual precipitation could be lost due to evapotranspiration worldwide.

However, in the Guder Catchment, between 7% and 9% of annual precipitation could be lost under both future scenarios. Land use types in a region can alter meteorological conditions, affecting ET. The relationship among different land use patterns, meteorological conditions and ET lays the groundwork for understanding a region's hydrology and future research. There are several explanations for the variability in the water yield, including changes in land use/land cover, soil, surface rinse and soil water recharge [36]. In conjunction with these factors, the main factor in decreasing water yield in the watershed might be rainfall distribution and air temperature rise. The average monthly water yield shows that the average water yield generated by each hydrological response unit in the catchment is varied and could be lower due to lower rainfall in the dry season. Due to high precipitation, the water yield produced by HRU in the catchment could be very high during the rainy season.

Greenhouse gases are trapping more heat in the Earth's atmosphere, causing an increase in the global average air temperature that could result in a water yield decrement in the Guder Catchment. If humans continue to release greenhouse gases into the atmosphere at their current pace, the global average air temperature will rise. The Guder Catchment could experience an air temperature increment. Nevertheless, the rise will be smaller if we make significant changes, such as using more renewable energy instead of fossil fuels and managing the watershed through soil and water conservation and afforestation. Under RCP 8.5 in the 2080s, the Guder Catchment will experience the most significant precipitation reduction of up to 14.4% relative to the current or baseline precipitation of 1228 mm/year. According to the projection scenario of RCP, precipitation may decrease in the future concerning the current scenario. The assumption given by IPCC [4] for the temperature rise by 2100 relative to pre-industrial temperatures given by the RCP 8.5 pathway may impact the water balance components. The change in precipitation may be due to the air temperature change, which could affect the whole hydrological cycle. Factors such as mountain ranges and prevailing winds may affect precipitation characteristics, and land-use change may impact the Guder Catchment. In this study, future changes in land use were assumed to be constant to examine the effects of climate variables on change, keeping all other factors constant.

## 6. Conclusions

Precipitation, evapotranspiration, surface runoff, groundwater flow, lateral flow and water yield are all hydrological components that could be influenced by human activities that cause climate change. Understanding hydrological cycles in the watershed and knowing the water yield potential of the basin help to plan and manage the water in the area properly and to evaluate water productivity. Due to the limited number of meteorological and hydrological stations in the region, the lack of critical climatic and flow data in the watershed exacerbates the problems in assessing hydrology in the Guder watershed. However, with the existing knowledge in data-scarce regions, the prediction of the main hydrological processes using semi-distributed hydrological models is vital for water resource management. The SWAT model was used to determine the Guder Catchment's hydrological parameters. The SUFI-2 methodology of the SWAT model was used to calibrate and validate the hydrological modeling of the Guder Catchment, and then water balance estimation was performed.

With climate change becoming a more significant hazard, significant adverse effects on water sources will probably occur in the future. The water yield is a water balance component that tells us how much water can be produced from a catchment or watershed usable by the community for socio-economic development or discharged from the watershed. Proper watershed management and awareness are critical to resolving water resource concerns through climate change in the Guder Catchment. In general, the current study provides an early estimate of the water yield and water balance at the basin level in the Guder Catchment. The overall water yield may be reduced by 3.2 mm per year under RCP 8.5 from 2024 to 2086, and a notable decreasing trend was detected. The regional governing

authority will prioritize initiatives to solve the community's water-related challenges based on this computation. Therefore, studies of the current water balances and current and future water yield may provide a more concrete basis for planning water projects and management. These findings can be utilized to conduct additional research to identify the effects, changes in the land cover over time and climate change implications under various scenarios.

**Author Contributions:** Conceptualization, analysis, writing—original draft preparation, review, and editing: T.M.G.; methodology, analysis, review and editing: H.Z., S.B., C.Y., Y.L., F.L. and H.L. All authors have read and agreed to the published version of the manuscript.

**Funding:** This study was supported by key R & D project funding from Jilin Province Science and Technology Department, China, 20200403070SF.

**Institutional Review Board Statement:** Not applicable.

**Informed Consent Statement:** Not applicable.

**Data Availability Statement:** Not applicable.

**Acknowledgments:** The authors would like to thank the Ethiopian Ministry of Water, Irrigation, and Electricity for the data they provided and all friends, colleagues and advisors for their encouragement.

**Conflicts of Interest:** The authors declare no conflict of interest.

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
