# Peer review of "Estimation of Hydrological Components under Current and Future Climate Scenarios in Guder Catchment, Upper Abbay Basin, Ethiopia, Using the SWAT"

_sustainability, doi:10.3390/su13179689_

Round 1
Reviewer 1 Report
The authors used Soil and Water Assessment Tool (SWAT) model to simulate the hydrological fluxes in the upper Nile Basin by considering current and future climate conditions. As the Nile water is very critical from integrated water resource management, such kind of analysis are important to adequately address issues among stakeholder, policy makers and the scientific community. Highlighting its importance not only from academic perspectives but also from socio-economic perspective.
The SWAT model is adequately calibrated and validated based on observation data using SWMM coupled optimization tools. The authors indicated the existence of simulated and statically significant trends of variability in terms of water yield.
General comment: Good to show the hydrological modeling using SWAT in your cases studies in Muger watershed, in Upper Nile Basin. Considering the fact there exists multiple hydrological, ecological, hydrogeological and environmental studies in upper Nile watershed, you should clearly indicate the uniqueness of this study, specially to support the scientific community.
Specific comment
Line 59: Delete unnecessary details, e.g. “Classification of hydrologic geographies and systems with the procedure of small-59 scale physical models, mathematical analogs, and computer simulations can be referred 60 to as Hydrological Modeling [14] and hydrological processes can be mathematically rep-61 resented by Hydrological Model [15], and”
Line 86: The data and method section should be explicitly discussed, if possible, separately. Discuss the data and the study area (model discretization and parameterization etc) and another section for methodology.
Line 269: Very good data presentation the monthly precipitation (what are the line indicated around the median inside the box and whisker plot?
Line 204: In the downscaling section, how accurate is the large-scale gridded data downgraded to Muger level watershed? How do you quantify any discrepancy associated with downscaling related errors?
Line 313: How many gauges station is used to calibrate and validate the SWAT model?
Line 324: In the sensitive analysis stable, it is not convenient to understand the parameter name easily. Rather than using the SWAT default abbreviation, it is better to use the respective parameter for reader to understand it.
Author Response
Dear Editor and Reviewer of Sustainability journal, This research was conducted to estimate water balance components and water yields under current and future climate change scenarios and trends in the Guder Catchment of Upper Blue Nile, Ethiopia, using the Soil and Water Assessment Tool (SWAT). Thank you very much for your professional advice. The author has adjusted the original considerably and reacted one by one based on feedback from the reviewers. These revisions not only improved the original article, but they also led the author to think about the science problem in greater depth, so thank you so much!
Note: The extensive editing of the English language has been done using PREMIUM GRAMMARLY SOFTWARE.
The correction has been done according to the suggestion given by the reviewers. Please see the file attached.

Reviewer 2 Report
After careful review of the current manuscript " Estimation of Hydrological Components under Current and future climate scenarios in Guder Catchment, Upper Abbay Basin, Ethiopia, Using SWAT" here are my comments:
-Please check SWAT is semi distributed model or lumped
-Authors only calibrated the SWAT hydrological model on monthly basis, why?
-Why not used the recent data for calibration and validation, because after 2001, may be some development occur in catchment. Please use recent data for calibration and validation of model.
-please add the table of best fit values of parameters used in SWAT model
- In table6 , why ET values in basin are very low? and also write mean instead of "main"
-Figure8, caption should be like Mean monthly water yield
Author Response
Dear Editor and Reviewer of Sustainability journal, This research was conducted to estimate water balance components and water yields under current and future climate change scenarios and trends in the Guder Catchment of Upper Blue Nile, Ethiopia, using the Soil and Water Assessment Tool (SWAT). Thank you very much for your professional advice. The author has adjusted the original considerably and reacted one by one based on feedback from the reviewers. These revisions not only improved the original article, but they also led the author to think about the science problem in greater depth, so thank you so much!
Note: The extensive editing of the English language has been done using PREMIUM GRAMMARLY SOFTWARE.
The correction has been done according to the suggestion given by the reviewers. And the correction/the response has been attached.

Round 2
Reviewer 1 Report
Thank you for addressing my comments.
This manuscript is a resubmission of an earlier submission. The following is a list of the peer review reports and author responses from that submission.
Round 1
Reviewer 1 Report
The study uses SWAT hydrological model to perform hydrological modeling in Guder, part of Nile Basin in Ethiopia. They authors used multiple data to formulate the hydrological conceptual model and multiple optimizations, calibration, validation, and sensitive analysis was done. The authors use statistical and model output to discuss the hydrological condition in the study area. The good point of the study is to uses multiple data set to develop the model and undergoing model validation to check model predictive capability. The weakness of work is poor introduction and flow to identify the research gap, objective, and model implication in real life.
Specific:
Line 85: The objective should relate to the research gap. Calibration and sensitive analysis cannot be your primary objective.
Line 282: Consistent figure captions (Fig. 3 and Fig. 4)
Line 321: What is the unit of Table 5? (Runoff and rainfall)
Line 342: How does the Sen’s slope value for each parameter in Table 6 related and discuss in detail in its impact in the study area from hydrological system understanding perspective.
Line 348: Table 7 has groundwater component output from the model. How do you verify the model output with respect to the groundwater flux model output in Table 7?
Reviewer 2 Report
Dear Authors,
The manuscript covers present and future changes in discharge components in Guder catchment in Ethiopia. Research is timely and should be published. However, the manuscript requires corrections. Therefore, I recommend major revisions prior its acceptance.
Major comments:
1) Introduction should be framed with a wider context related to climate change (i.e., refer to IPCC reports dealing with water scarcity, food production etc.). In some parts, there is too much of theoretical description of hydrological modelling (lines 51-56, 70-73). There is too long description of SWAT model without referring to studies in Ethiopia or other similar regions where this model was used.
2) In Method sections, there is a lack of information on discharge data range used for model calibration and validation. Line 133-134, there is just vague information of using the MoWIE dataset without even referring to database. In subsection 2.4 description of SWAT model calculations should be focused only on the calculation used by Authors in the manuscript.
3) In data presentation, there is missing plot (e.g., box plots) showing air temperatures during analysed period. In results section, there should be some rearranging as Table 7 showing hydrological data comes at the end of section. Also, it is confusing when there are few terms used for runoff (e.g., water yields, WYLD). This should be clarified in the manuscript.
4) In discussion, the last paragraph is too general (global climate change) and should be focused on forecast of discharge in Gudar catchment and how this can tell us about changes in similar environments. In conclusions, there is a similar problem as in the last paragraph of discussion – lines 463-469 are far too general.
Detailed comments:
L14 – what do you mean by yields? It should be specific runoff. This should be corrected in entire manuscript.
L102-104 – Mean air temperature are required along with reference to meteorological data, from where Authors obtained them.
L104 – lowest not lowermost
L131 – Air temperature not temperature should be used. Also in other parts of the manuscript.
L133 – stream not steam
L212 – term alienated should be altered.
L222 – validation
L270 – there should be R2
L342 – Mk should be defined
L409 – discussion not discussions
Figure 1 – Location of study area should be shown on map of Africa. There is missing information on main cities, rivers, mountains in the map of Ethiopia. There are three figures, which should be mentioned in figure caption.
Figure 2B – there is lack of explanation in legend.
Figure 3 – date on X-axis should be corrected.
Figure 4 – it is recommended that minimal and maximal total precipitation will be shown (e.g. box plot)
Figure 5 – shows specific runoff (in mm), while runoff unit is for example m3. It is recommended that minimal and maximal total precipitation will be shown (e.g. box plot)
Figure 6 – air temperature shown as line will be beneficial for the plot.
Figure 7 – this is specific runoff not runoff. Total precipitation will be appropriate to show in plot.
Figure 8 – In the southern part of the basin, altitude exceeds 3000 m a.s.l. so why precipitations are the lowest? This potential error on precipitation quantification may have affected hydrological modelling.
Figure 10 – colours of bar should be explained as this is another figures as compared with Figure 9.
Figure 11 – both figures can be placed in one graph.
Best regards,
Reviewer.
Reviewer 3 Report
The work was carried out using well-known approaches and methods presented in almost a standard form in many similar papers, including papers from Ethiopia.
After reading the manuscript, I had severe doubts about the advisability of its publication for two main reasons.
- The authors did not provide any relevant hydrological data based on which they carried out the modeling. What data were used to calibrate and validate the model (Fig. 3)? What river is it? Where is the analyzed hydrological station located? What is the temporal resolution of the hydrological data? Why does the text say about some reservoir (lines 116-121)? What is its role in your research? What is this mysterious method about which we know nothing from the manuscript? If authors have average annual and monthly average hydrological data, they should provide them in the Appendix or Supplementary Materials.
- Table 7. The results of this table shocked me. How did it happen that in summer (June to August), when there should be the highest evaporation due to high temperatures and the highest precipitation, the share of evapotranspiration (ET) was only 5-6% of rainfall, and the rest (94-95%) is runoff!? Nonsense! Even in the colder northern regions of Europe, this is rarely achieved. On the windward slopes of the Ethiopian Highlands, the total runoff coefficient does not exceed 0.3 (30%). Moreover, why was the groundwater runoff several times greater than the surface runoff, given that the dominant landscape in the studied basin is agricultural land (Fig. 2b)? Other things being equal, cultivated land favors more intensive surface runoff and increased soil erosion. All this has been observed in Ethiopia for many decades. Additionally, less evaporation (according to your calculations) should have favored a more significant surface runoff. Therefore, I don’t believe all of your simulated results. Furthermore, the elements of the water balance are not confirmed in any way by real river water discharge and independent methods. In this regard, how can you model something for decades to come?
From all of the above, I do not see either the scientific or the practical significance of this research in its current form.